# Less and Less Noble: Local Adsorption Properties of Supported Au, Ni, and Pt Nanoparticles

**DOI:** 10.3390/nano13081365

**Published:** 2023-04-14

**Authors:** Andrey K. Gatin, Sergey Y. Sarvadii, Nadezhda V. Dokhlikova, Sergey A. Ozerin, Vasiliy A. Kharitonov, Dinara Baimukhambetova, Maxim V. Grishin

**Affiliations:** 1N.N. Semenov Federal Research Center for Chemical Physics of Russian Academy of Sciences (FRCCP RAS), Kosygina Street 4, 119991 Moscow, Russia; 2Moscow Institute of Physics and Technology, National Research University (MIPT), Institutskiy Pereulok 9, 141701 Dolgoprudny, Russia

**Keywords:** nanoparticles, gold, nickel, platinum, carbon, interface, STM, chemical bonding, chemisorption, adsorption complex

## Abstract

In this work, we studied the local adsorption properties of gold, nickel, and platinum nanoparticles. A correlation was established between the chemical properties of massive and nanosized particles of these metals. The formation of a stable adsorption complex M-A_ads_ on the nanoparticles’ surface was described. It was shown that the difference in local adsorption properties is caused by specific contributions of nanoparticle charging, the deformation of its atomic lattice near the M–C interface, and the hybridization of the surface s- and p-states. The contribution of each factor to the formation of the M-A_ads_ chemical bond was described in terms of the Newns–Anderson chemisorption model.

## 1. Introduction

Despite many years of research, the mechanisms of catalytic reactions on surfaces remain largely unclear [1]. This is especially true for catalytically active nanostructured systems. Due to the fine interaction of various components of such systems and a number of quantum size effects, their catalytic activity can change significantly [2,3].

In the work of Hammer and Nørskov [4], an attempt was made to consider the chemical activity of gold, nickel, platinum, and copper in terms of chemical bond formation in M-A_ads_ adsorption complexes. The model they propose effectively describes the formation of stable complexes on the surfaces of bulk metals. However, it is impossible to use similar regularities to directly describe the chemical activity of nanosystems. Indeed, the surface of one nanoparticle can have regions with different adsorption properties [3,5].

It is extremely difficult to model such systems. A particle with a size of several nanometers exceeds to a significant degree the distances that are characteristic of surface migration or interaction between molecules. At the same time, the particle interacts with the bulk substrate. The need to model subsystems of different scales further muddles calculations that are already complicated. As a result, the only thing we know is that we know nothing [6]. All observed regularities of dissociation, adsorption, migration, and desorption are the result of a fine interplay of various effects with different local contributions [7,8,9,10,11].

The question then arises: is it possible to carry out an experiment to determine these effects and indicate exactly which of them will prevail in one or another region over the nanoparticle’s surface? Of course, such experiments are almost an art form and require experimenters with the highest skill level. To such experiments is our work devoted.

Our research group studied the adsorption of various gas-phase reagents on the surface of nanostructured systems based on gold, nickel, and platinum. The considered systems are very complicated. On one hand, ab initio quantum-chemical modeling is very complicated in this case. On the other hand, the chemical properties of materials have changed significantly and cannot be strictly described in terms of solid-state physics. Comparing our STM/STS results with the calculations of the Nørskov group [4], we managed to describe the observed spatial regularities in terms of the formation of stable adsorption complexes on the surfaces of nanoparticles. As a result, we have established a correlation between the chemical properties of massive and nanosized particles of these metals. This approach eliminates the problem of describing such systems. Without loss of generality, it can be applied to similar systems based on nanoparticles of both transition and non-transition metals.

## 2. Experimental

Nanoparticles were synthesized using the impregnation–precipitation method on the surface of chemically inert highly oriented pyrolytic graphite (HOPG, AIST-NT, Moscow, Russia) with an angular spread of c-crystallite axes of 0.8° [12]. Aqueous solutions of chloroauric acid H[AuCl_4_], nickel nitrate Ni(NO_3_)_2_, and chloroplatinic acid H_2_[PtCl_6_] with a metal concentration of 2–2.5 mg/L were used as precursors. The precursor solution was applied to the cleaned HOPG surface, which looked like vast atomically smooth C(0001) terraces. After drying the solution, the samples were placed in an STM chamber, where they were calcined under ultrahigh vacuum (UHV) conditions at a temperature of 700 K for 28–30 h.

During calcination, the precursor solution H[AuCl_4_] decomposed with gold nanoparticle formation as follows [13]:(1)$$2{\mathrm{HAuCl}}_{4}\cdot 3H_{2}O\overset{345–450\;K}{\to }{\mathrm{Au}}_{2}{\mathrm{Cl}}_{6}+2\mathrm{HCl}+6H_{2}O$$
(2)$${\mathrm{HAuCl}}_{4}\cdot 3H_{2}O\overset{345–450\;K}{\to }{\mathrm{HAuCl}}_{2}{\left(\mathrm{OH}\right)}_{2}+2\mathrm{HCl}+{\mathrm{Cl}}_{2}+H_{2}O$$
(3)$${\mathrm{Au}}_{2}{\mathrm{Cl}}_{6}\overset{450–505\;K}{\to }2\mathrm{AuCl}+2{\mathrm{Cl}}_{2},$$
(4)$$4{\mathrm{HAuCl}}_{2}{\left(\mathrm{OH}\right)}_{2}\overset{450–505\;K}{\to }4\mathrm{Au}+8\mathrm{HCl}+2H_{2}O+3O_{2},$$
(5)$$2\mathrm{AuCl}\overset{505–570\;K}{\to }2\mathrm{Au}+{\mathrm{Cl}}_{2}$$

For the formation of platinum nanoparticles, the precursor must be calcined in an H_2_ atmosphere. The reaction occurs according to the following scheme [14]:(6)$$H_{2}{\mathrm{PtCl}}_{6}\cdot 6H_{2}O\overset{545–570\;K}{\to }{\mathrm{PtCl}}_{4}+2\mathrm{HCl}+6H_{2}O,$$
(7)$${\mathrm{PtCl}}_{4}+2H_{2}\overset{520-570\;K}{\to }\mathrm{Pt}+4\mathrm{HCl}.$$

Nickel nanoparticles are formed in a similar way upon calcination in H_2_ [15]:(8)$$\mathrm{Ni}{\left({\mathrm{NO}}_{3}\right)}_{2}\overset{\sim 573\;K}{\to }\mathrm{NiO}+O_{2}+{\mathrm{NO}}_{2},$$
(9)$$\mathrm{NiO}+H_{2}\overset{\sim 535\;K}{\to }\mathrm{Ni}+H_{2}O.$$

Thus, the given calcination conditions provide the complete decomposition of all precursors and the formation of metal nanoparticles on the HOPG surface.

All experiments were carried out in a setup consisting of an ultrahigh vacuum chamber, a scanning tunneling microscope (UHV VT STM, Omicron NanoTechnology, Taunusstein, Germany), a quadrupole mass spectrometer (Hiden Analytical Limited, Warrington, UK), a gas-pumping system, the pipelines for gas injection, and other auxiliary equipment for sample manipulation. The residual pressure in the UHV chamber did not exceed 10^−10^ mbar. This setting prevents uncontrolled changes in the chemical composition of the samples due to residual gases and provides an unambiguous interpretation of the results obtained.

In the experiments, we used tungsten STM tips produced by electrochemical etching and cleaned by argon-ion sputtering under UHV conditions. Only those tips were used that demonstrated a reproducible S-shaped curve of the volt–ampere characteristic (VAC) while scanning a clean HOPG surface. This shape of the VAC curve is typical for metal–metal tunnel nanocontacts [16].

The gas composition in the UHV setup was controlled by mass spectrometry at all stages of the experiment, including the injection of O_2_, H_2_, and CO of ultra-high purity. During the experiments, the pressure of the gas reagents was 1.33 × 10^−6^ mbar at a temperature of 293 K. The sample exposure was measured in Langmuir (1 L = 1.33 × 10^−6^ mbar·s). After the gases were pumped out from the UHV chamber, the synthesized samples were probed using STM/STS methods.

## 3. Gold

Consider the gold-based nanostructured system. This was the first system we studied. Since we managed to observe hydrogen adsorption and some chemical reactions over the surface of gold nanoparticles, many questions have arisen about this system [17,18,19].

In all our experiments, the structure of the synthesized coating was practically the same. Gold nanoparticles mainly cover the edges of graphite terraces and decorate defects on the substrate surface, but there are also single nanoparticles on smooth graphite areas (see Figure 1a). The particles have an average lateral size of about 5 nm, and their height is 1.5–2 nm above the atomically smooth graphite surface (see Figure 1c,d). 

The results of the STS measurements show that the nanoparticles possess a metallic electronic structure (see Figure 1b). The VAC curves of the tunnel nanocontact have no zero-current section.

We do not discuss in detail the experiments with gold nanoparticles and various gas reagents, as such discussions can be found in our previous works [17,20]. We only want to note the high reproducibility of the results obtained.

With the low exposure of gold nanoparticles to H_2_ (200 L), we observe a change in their electronic structure near the Au–C interface (see Figure 2). On the periphery of the nanoparticle, an annular region is formed with an electronic structure of the semiconductor type.

At high levels of exposure to H_2_ (2000 L), the formation of a semiconductor layer was observed over the entire surface of the nanoparticle [17]. We also observed the inhibition of hydrogen adsorption over the surface of gold nanoparticles due to their positive charge [18].

However, in this case, the formation of the annular region posed a serious challenge to us. There is no doubt that nanoparticles’ charges affect the dissociation of H_2_ molecules on gold and the formation of stable Au-H [21,22]. But why is the surface distribution of Au-H so uneven? In our previous works, we considered the possibility of the formation of such a complex [20], but the locality of this process cannot be explained by charge effects alone. After all, the charging of a supported nanoparticle due to the difference in the electron work functions is too global a phenomenon.

In our previous work, we noted that the result of quantum chemical modeling was paradoxical [22]. According to the calculation, a regular cluster of gold atoms with a fixed structure located over a single graphene layer is positively charged, although the opposite is experimentally observed. Only when taking into account the rearrangement of the cluster geometry and the optimization of its atomic structure did we obtain a result that contradicted neither experiment nor common sense.

Of course, within the simulation, the separation of the contributions of charge and geometry is artificial in the case of a small gold cluster. In fact, these two factors mutually influence each other. However, in the experiment, such a separation is not required. It is sufficient that the contribution of each factor will vary locally. As a result, one can see that the adsorption complex is localized on certain areas of the nanoparticle’s surface, and that the surface distribution will be strict if the experimenter is lucky enough.

Consider the process of the formation of the Au-H chemical bond under the assumption that charging is a more global factor, and the geometry factor actively affects the adsorption properties only near the Au–C interface.

Considering the electronic structure of gold, one can see that its d-band is deeply located (see Figure 3a). The interaction of the adatom with the d-band states leads to the formation of bonding and antibonding orbitals, both located below the Fermi level. Since the antibonding contribution always exceeds the bonding one, the formation of a stable adsorption complex does not occur. This explains the chemical inertness of bulk gold [4]. 

The situation changes if we consider nanostructured gold. Against the backdrop of the d-band states, the surface states begin to manifest themselves clearly (see Figure 4a) [20]. These surface sp-states contribute significantly to M-A_ads_ chemical bonding and lead to the formation of occupied σ_spa_ and empty σ*_spa_ orbitals. However, the occupation of the bonding σ_spa_ orbital can be changed. Such an experiment can be organized in a broken tunnel junction, when an STM tip and a supported nanoparticle form an asymmetric capacitor. In such a system, one can efficiently change the occupation of the d-band and the surface sp-states. To achieve this, one must simply change the value or polarity of the voltage applied to the vacuum gap. We have demonstrated this in our previous work [18].

A nanoparticle can also obtain an excessive charge simply due to an interaction with the substrate [23]. A nanoparticle will be negatively charged if its work function exceeds the one of the support, and so the sp-states will be occupied with excessive electrons (see Figure 4b). However, charge transfer in such a system is limited. No one knows whether the difference between the work functions of gold and graphite is sufficient for the effective occupation of the surface sp-states and for the down-shift of the center of the sp-band below the Fermi level.

Apparently, this is not enough. The center of the hybrid sp-band is most likely to remain above the Fermi level when the gold cluster is charged due to interaction with the support. The excessive charge doesn’t exceed 5e [24]. Of course, one cannot exclude the formation of Au–H complexes at defects of nanoparticles surface. That is, adsorption can occur at single points, but it is not widespread. According to the quantum chemical simulation, for a regular gold cluster interacting with a single layer of graphene, the formation of a stable Au-H bond occurs only in the case of the deformation of the gold cluster [22].

As such, from the charging effect, we can move on to considering the influence of the atomic structure. A change in the crystal lattice parameters can lead to a change in the d-band width and down-shifting of its center [25]. This seems to be exactly what happens near the Au–C interface (see Figure 4c). The deformation of gold deposited on graphite is described in some other works [26]. In this case, the lattice is compressed, the orbital overlap increases, and the d-band center shifts downward. The same happens with surface states. This means that the lattice compression results in the efficient occupation of higher states. Together with charging, this leads to a noticeable downward shift of the center of the hybrid sp-band to below the Fermi level (see Figure 4d). The antibonding orbital σ*_spa_ remains empty in this case. As a result, we observe the formation of a stable Au–H complex in the region of lattice compression. Thus, the observed annular structures can be interpreted as regions where the surface sp-states are efficiently occupied.

In the central part of the nanoparticle, the occupation of the sp-states is influenced only by the excessive charge. In the interface region, rearrangement of the atomic structure makes an additional contribution. As a result, the formation of a stable Au–H complex occurs only at the periphery of the nanoparticle.

This model leaves open the question of how the nanostructured Au–H system will be rearranged with increasing exposure. Indeed, we observe the complete coverage of the nanoparticle at high levels of exposure to H_2_. Moreover, the interaction of surface sp-states with an adatom also requires more careful consideration: even if the center of the hybrid sp-band does not contribute, slightly lower occupied states may participate in bond formation. Nevertheless, this model effectively explains the observed surface distribution of the adsorbate. It is in good agreement with the results of quantum chemical calculations and experiments with an external field [18]. It allows us to describe qualitatively analogous patterns for nickel and platinum nanoparticles, which we now consider in more detail.

## 4. Nickel

As mentioned above, nickel nanoparticles were synthesized from nickel nitrate salts. After all stages of drying, calcination, and reduction in H_2_, nanoparticles formed on the HOPG surface. The nanoparticles are mainly located at the edges of graphite terraces and decorate defects, as in the case of gold. There are both agglomerates and individual nanoparticles on the surface of graphite (see Figure 5a). According to the results of STM measurements, the particles have an average lateral size of 5–6 nm, and their height is about 1–2 nm over the HOPG surface (see Figure 5c,d). According to the shape of the VAC curves, the conductivity of the nanoparticles is of the metallic type (see Figure 5b). This suggests that calcination of the sample in H_2_ at a high temperature leads to the reduction of nanoparticles or at least to the deep metallization of their surface.

As a result of exposure to O_2_ (200 L), an oxide layer is formed on the surface of the nanoparticles. Its band gap is 1.1–1.8 eV, and its surface distribution demonstrates inverted but still analogous annular regularities, as in the case of gold (see Figure 6). The oxide is formed only in the area that is maximally far from the Ni–C interface, that is, at the top of the nanoparticle. A region is formed along the perimeter where the nanoparticle retains the former electronic structure of the metallic type.

The experiment completely reproduces the results of our previous work, where we assumed that this locality of oxide formation is due to the excessive charge of the nanoparticle and its influence on oxygen diffusion [27]. However, several factors are most likely to work here, as in the case of gold.

Let us consider the formation of the chemical bond in the Ni–O complex, taking into account two factors: charging and the lattice deformation of the nanoparticles due to interaction with the support. Compared to gold, nickel has a high d-band (see Figure 3c). Its upper edge is slightly above the Fermi level. Such a d-band position facilitates the adatom bonding to the surface, since the antibonding σ*_da_ orbital is unoccupied. At the same time, the surface s-states hardly contribute to bonding. All this leads to the formation of a stable Ni–O complex on the surface of bulk nickel [4].

According to various data, the electron work function for nickel is 5.04–5.35 eV [28]. When deposited on graphite, nickel acquires an excessive negative charge. As a result, free d-band states are occupied, and its center is down-shifted. Is it sufficient to effective down-shift the σ*_da_ orbital below the Fermi level? The antibonding orbital is most likely to remain above the Fermi level, since the charge influence is not crucial (see Figure 7b). The oxide formation is observed experimentally at the top of a nanoparticle, which is consistent with our assumption. In other words, in the absence of any other factors, nanosized nickel retains the chemical properties inherent to bulk nickel.

Let us now consider the effect of deformation. The nickel lattice is dilated when deposited on most metal and oxide supports [25]. However, when nickel is deposited on some graphite-like supports, its lattice is compressed [29,30]. We can suppose that the same result is observed in our experiments. That means the d-band will become narrow and occupied, and, again, its center will shift down (see Figure 7c). The lattice compression is most likely to provide efficient occupation of the σ*_da_ orbital (see Figure 7d).

Therefore, both factors considered by us lead to a down-shift of the d-band center and a decrease in the Ni-O binding energy. Since the surface migration of oxygen is prevented due to the intensive penetration of O atoms under the surface layer of Ni atoms [31,32], the annular region with a metallic structure correlates precisely to the effective occupation of the antibonding orbital σ*_da_.

Of course, one should consider carefully the surface s-states available for occupation and their contribution to bond formation. However, this does not change the picture in general. There is no additional bonding to the adatom at the periphery, and a stable Ni–O complex is not formed.

## 5. Platinum

Let us consider the platinum-based nanostructured system. We found the regularities observed for it to be the most challenging. As mentioned above, platinum nanoparticles were obtained by the decomposition of chloroplatinic acid. The synthesized nanoparticles had a wide size distribution from 2 to 12 nm, and the distribution maximum was at 4–6 nm (see Figure 8c,d).

As in the two previous cases, the nanoparticles agglomerated on the defects of the support and decorated the edges of the HOPG terraces. Single nanoparticles were also present on the surface (see Figure 8a). STS experiments showed that the nanoparticles have a metallic electronic structure (see Figure 8b).

At a low exposure (20 L) to N_2_O, we observe changes in the electronic structure of the nanoparticle. One can distinguish the central and peripheral regions according to the STM results. The electronic structure of the metal type is retained at the top of the nanoparticle, but the VAC curves demonstrate a zero-current region close to the Pt–C interface. This means that the initial stages of oxidation are observed. This result correlates well with other works [33]. Therefore, we cannot declare the formation of a stoichiometric oxide, but this is sufficient to indicate that the adsorption complex is more stable at the periphery than in the central part.

For further experiments, freshly synthesized platinum nanoparticles were exposed to oxygen (2000 L) at 700 K. We achieved the formation of a Pt–O adsorption complex over the entire nanoparticle surface. The observed band gap corresponded to the reference one [33].

Afterwards, we tried to reduce the platinum. A short exposure to H_2_ (200 L) led to the same annular regularities we observed. One can distinguish the central and peripheral regions of the nanoparticle surface (see Figure 9a). The central part is reduced, and the periphery retains the semiconductor electronic structure (see Figure 9b). We observed the same effect after the exposure of fully oxidized nanoparticles to CO (200 L). Since the observed regularities do not depend on the reducing agent, we can state that they are definitely due to the stability of the Pt-O, and not to the specifics of the chemical reaction. This result is in good agreement with the results obtained for platinum monocrystals [33].

Comparing the adsorption properties of platinum with those of gold and nickel described above, one can see that platinum nanoparticles look more like gold than nickel: a stable M-A_ads_ adsorption complex is formed at the periphery of the nanoparticle close to the M–C interface (see Table 1).

This is a very interesting result. Indeed, if we consider the formation of the M-A_ads_ adsorption complex, taking into account the charging of a nanoparticle and the deformation of its atomic layers close to the M–C interface, we can see that we should have obtained the exact opposite result.

The location of the platinum d-band is like that observed in the case of nickel. Its center is close to the Fermi level and it is partially occupied (see Figure 3b). The difference in work function and the compression of the platinum atomic lattice should lead to the effective occupation of free electronic states [28,34]. As such, the chemical bond M-A_ads_ should weaken at the periphery of the platinum nanoparticle due to its interaction with the HOPG support. The platinum nanoparticle should interact with O adatoms, as did nickel (see Figure 10a–d).

The experiment contradicts our expectations. This indicates the presence of another factor not taken into account so far. Thus, it is necessary to find a factor that makes the electronic structure of platinum nanoparticles more gold-like.

An increase in the contribution of the s- and p-states can lead to stronger bonding of the adatom with the surface. This is exactly what we observe in the case of gold: the sp-state’s contribution becomes significant when σ*_da_ is already occupied and effective occupation of σ_spa_ is just beginning. In the case of nickel, the positions of the d- and s-bands’ centers do not differ greatly. Therefore, the antibonding σ*_da_ and σ*_sa_ are occupied or emptied almost simultaneously. However, if we could somehow effectively set apart the centers of the d- and s-bands, it would increase the difference between σ*_da_ and σ*_sa_ and the electronic structure of nickel would become more gold-like.

We have not yet considered the cohesive energy. Hammer and Nørskov’s research group mention this factor, but do not discuss it in detail [4]. Considering the hypothesized surface s-band for gold, one can see that the hybridization and formation of the sp-band led to the shift of the pure s-band’s center. Since its center differs significantly in energy from the d-band center, the hybridization of the s- and p-states has practically no effect on the chemical properties of gold nanoparticles, but, in the case of platinum, even a small shift of the s-band center may become significant (see Figure 10e–h). One can say that sp-hybridization truly makes platinum gold-like.

This assumption is also supported by the fact that the hybridization contribution will depend on the particle size [20]. Indeed, for large particles, the surface s- and p-states become insignificant against the background of the d-band. Nanoscale platinum will react with gases as gold nanoparticles do, while bulk platinum appears to be nickel-like in its adsorption properties (see Table 1). A similar result was obtained by quantum-chemical modeling [35]. In the case of a platinum cluster interacting with a graphite flake, the bonding with the O adatom weakens with distance from the M–C interface.

## 6. A Quick Glance at the Adsorption Mechanism

Thus, we have considered the regularities of the M-A_ads_ complex formation for gold, nickel, and platinum nanostructured systems. Now, we briefly discuss the other stages of adsorption to provide more information about the interaction of gas molecules with nanoparticle surfaces. This process includes molecular physisorption and migration, the dissociation and formation of a stable M-A_ads_ complex, and the surface migration of the adsorption complex and its desorption (see Figure 11). The main problem is that we cannot watch these stages in dynamics, since the STM/STS methods allow us only to compare the initial and final conditions of the investigated system. However, this is sufficient to make some assumptions about all the stages of adsorption. 

There is little to say about two first stages—molecular physisorption and migration. For hydrogen adsorption on gold nanoparticles, these stages are quickly followed by chemisorption [17,18]. The same takes place for oxygen adsorption on nickel nanoparticles, which leads to oxide formation [27]. Only in the case of oxygen adsorption on platinum nanoparticles did we manage to observe some chaotic changes of the VAC curves. However, just a few minutes after the exposure, the system returned back to a stable condition with a pure metal electronic structure.

Hydrogen chemisorption on gold nanoparticles occurs only in a dissociative form. The antibonding σ* orbital interacting with the metal surface is shifted below the Fermi level and occupied by electrons tunneling from the nanoparticle, and so the H–H bond is weakened [18]. This resonant chemisorption mechanism seems also to be valid in the case of O_2_ adsorption on platinum nanoparticles. Since the oxidation of platinum nanoparticles occurs only at T = 600 K, one can conclude that bond weakening is partial—maybe with the formation of charged O-O species—and additional energy is necessary for complete dissociation (see *E*^a^_chem_ in Figure 11). This was proven in experiments with N_2_O, which dissociates at room temperature on platinum nanoparticles and is quickly followed by stable Pt-O formation. In the case of nickel nanoparticles, we did not manage to observe any intermediate forms of oxygen adsorption against the background of oxide formation [27].

Hydrogen dissociation on gold nanoparticles and the Au–H complex formation occur simultaneously. At the same time, the influence of the nanoparticle charge differs for these processes [18,22]. The dissociation probability increases with the negative charge, while the Au–H bond weakens. Complete dissociation inhibition with a positive charge has been demonstrated recently [18], but the effective negative charge should still be estimated for the complete inhibition of Au–H formation. For nickel and platinum nanoparticles, the question remains open, and some additional experiments should be carried out.

The formation of a stable Au–H complex does not occur at once. During the 24 h after the exposure to hydrogen, some relaxation processes take place in the nanostructured system, and the form of the VAC curves changes chaotically. This fact may point to the surface migration of chemisorbed H atoms and their transition to the bridge position from the upper and hollow ones [36]. The last two states are similar in energy, while the first one is 0.12 eV more beneficial (see Figure 11). One can conclude that the migration barrier should be low enough to provide a one-way H transition to the bridge position, which is stable at room temperature. Of course, the Au–H bonding energy is also influenced by the Au–C interface, as is demonstrated by the work presented.

Hydrogen adsorption does not lead to its dissolution in the gold nanoparticle lattice [19]. However, in the case of nickel, the adsorption of oxygen is followed by a significant rearrangement of the nanoparticles’ surface and the deep penetration of oxygen atoms into the lattice, as well as their diffusion and bulk oxide formation (see activation energies E^a^_pen_ and E^a^_dif_ in Figure 11) [27]. As such, the stability of the nickel-based nanostructured system is due to oxide formation. At the same time, the Pt–O complex tends to slow decomposition. At room temperature, pure platinum nanoparticles may be observed 3–4 weeks after their exposure to oxygen. One can conclude that oxygen adsorption is followed only by surface processes on platinum nanoparticles without deep penetration of oxygen atoms into the lattice. This fact allows us to estimate the desorption energy for gold- and platinum—based nanostructured systems. In the case of gold nanoparticles, the energy of H atom desorption exceeds 1.7 eV [36]; meanwhile, for O atoms on the surface of platinum nanoparticles, it is less than 1.1 eV (see E_des_ in Figure 11).

It is clear that many questions remain open. Charge and lattice distortion may seem to add new pieces to this puzzle, but the general regularities established—even for very different systems such as those of gold, nickel, and platinum—make this a solvable problem.

## 7. Conclusions

The studies carried out here made it possible to establish a correlation between the adsorption properties of bulk and nanosized particles of gold, nickel, and platinum. For all HOPG-supported nanoparticles, STM/STS experiments with low exposure to gas reagents showed a significant difference in adsorption properties at the top of the nanoparticle and in the region close to the M–C interface.

Such a difference in the adsorption properties is explained by various local contributions to the formation of a stable M-A_ads_ adsorption complex: from the charging of the nanoparticle, from the deformation of its atomic lattice close to the M–C interface, and from the hybridization of the s- and p-states.

The formation of a stable adsorption complex is considered on the surface of gold nanoparticles—a non-transition metal with an occupied d-band located far below the Fermi level. In this system, the formation of the M-A_ads_ chemical bond occurs due to the occupation of free sp-states. This process is caused by the electron transfer from the support to the nanoparticle and the compression of its atomic lattice close to the M–C interface. In this case, the occupied orbitals σ_da_, σ*_da_, and σ_spa_ are formed, while the antibonding σ*_spa_ remains unoccupied. The additional local contribution of the lattice deformation near the M–C interface increases the probability of adsorption, and this explains the formation of a stable Au–H complex at the periphery of the gold nanoparticles.

In the case of nickel—a transition metal—the nanoparticles’ d-band is not completely occupied. Its center is located close to the center of the s-band, and the antibonding σ*_da_ and σ*_sa_ orbitals are occupied or emptied almost simultaneously. Therefore, the M-A_ads_ binding energy decreases due to the effective occupation of the d- and s-bands caused by an excessive electron charge and the compression of the nanoparticle’s atomic lattice close to the M–C interface. Therefore, the formation of a stable Ni–O complex occurs at the top of the nickel nanoparticle, where the contribution of the lattice deformations is absent, and the contribution of negative charging is insufficient to prevent adsorption.

With an increase in the atomic mass, it is necessary to take into account the s- and p-states’ hybridization for transition metals. The d-band states contribute to the formation of the M-A_ads_ complex on the surface of bulk metal, leading to the formation of a bonding σ_da_ orbital. In this case, the contribution of the hybrid sp-band is insignificant, but the contribution of the s- and p-states increases for nanoparticles. The center of the hybrid sp-band is located above the center of the pure s-band. As a result, the electronic structure of the nanoparticles becomes similar to the structure of non-transition metals. Therefore, the adsorption properties of bulk platinum are similar to those of bulk nickel, while nanoparticles of platinum interact with adatoms such as nanosized gold. The formation of a stable Pt–O complex occurs at the periphery of platinum nanoparticles.

## Figures and Tables

**Figure 1 nanomaterials-13-01365-f001:**
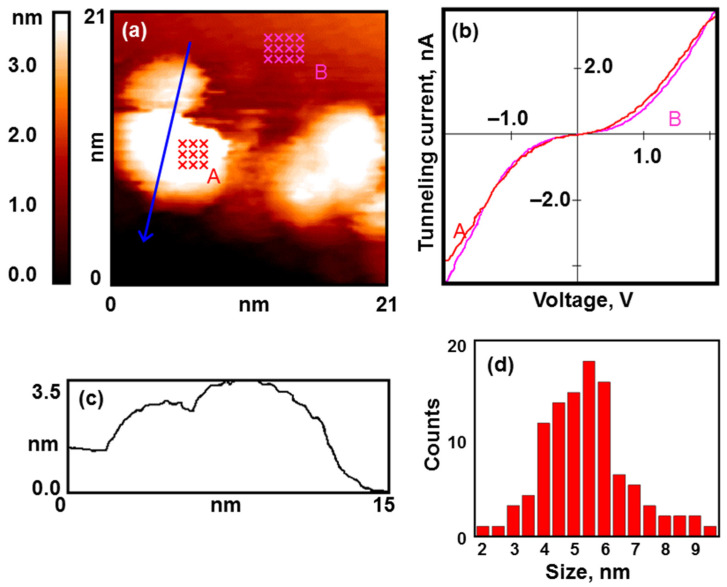
Nanoparticles of gold after calcination under UHV conditions. Results of the STM/STS measurement: (**a**) topography image of the HOPG surface with deposited gold nanoparticles; (**b**) VAC curves of the tunneling currents averaged over the set of points on the surface of the nanoparticles (red curve A) and HOPG (pink curve B) marked with crosses in (**a**); (**c**) profile of the surface along the cut line shown in (**a**); (**d**) size distribution.

**Figure 2 nanomaterials-13-01365-f002:**
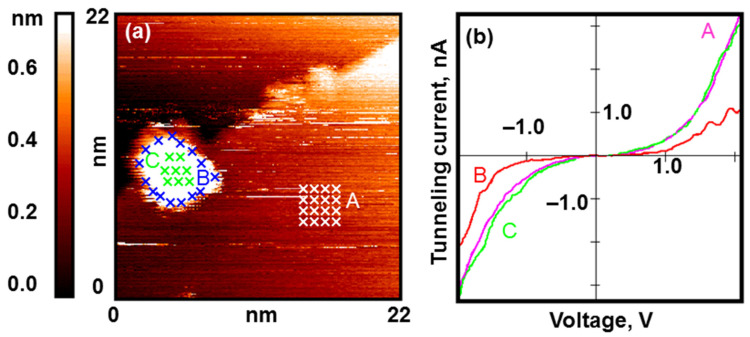
Gold nanoparticles after exposure to H_2_ (50 L). Results of the STM/STS measurement: (**a**) topography image of the HOPG surface with deposited gold nanoparticles with the points of the spectroscopy measurements marked with crosses; (**b**) VAC curves of the tunneling currents averaged over the set of points on the surface of the HOPG (pink curve A), central (green curve C), and peripheral (red curve B) areas of the nanoparticle marked in (**a**).

**Figure 3 nanomaterials-13-01365-f003:**
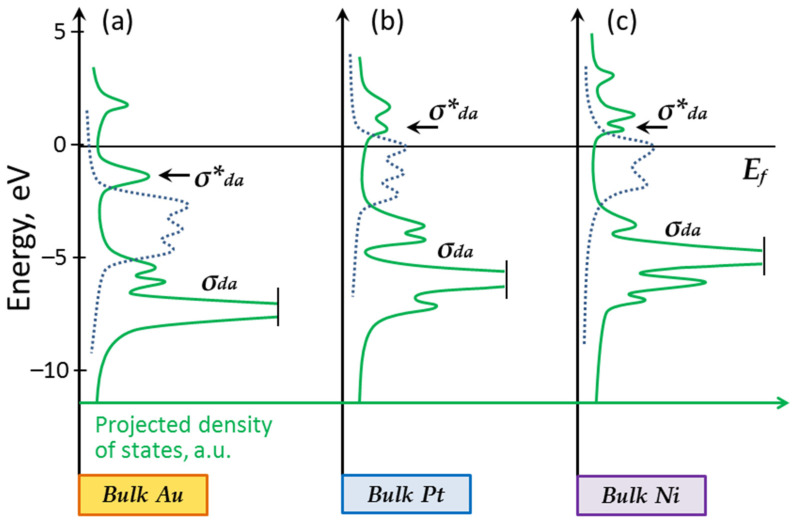
Interaction of adatom with d-band states of bulk gold (**a**), platinum (**b**) and nickel (**c**). Schematic diagram (compare with [4]). The density of states projected on adatom (green solid line) demonstrates resonances corresponding to the formation of bonding and antibonding orbitals. The density of the d-band states for the clean metal surface is shown for comparison in each case (blue dashed line).

**Figure 4 nanomaterials-13-01365-f004:**
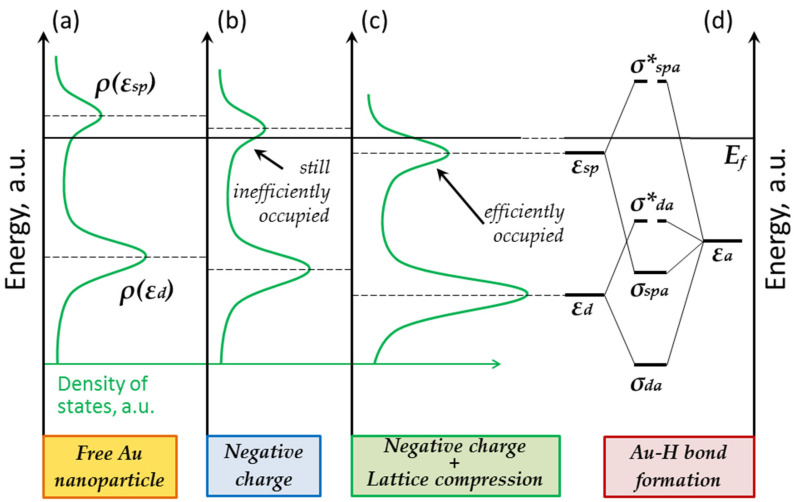
Influence of charge and deformation factors on the occupation of the surface sp-state and d-band state: (**a**) unsupported gold nanoparticle; (**b**) negatively charged gold nanoparticle; (**c**) gold nanoparticle with negative excessive charge and compressed atomic lattice; (**d**) formation of bonding and antibonding orbitals due to the interaction of adatom with sp-state and d-band state.

**Figure 5 nanomaterials-13-01365-f005:**
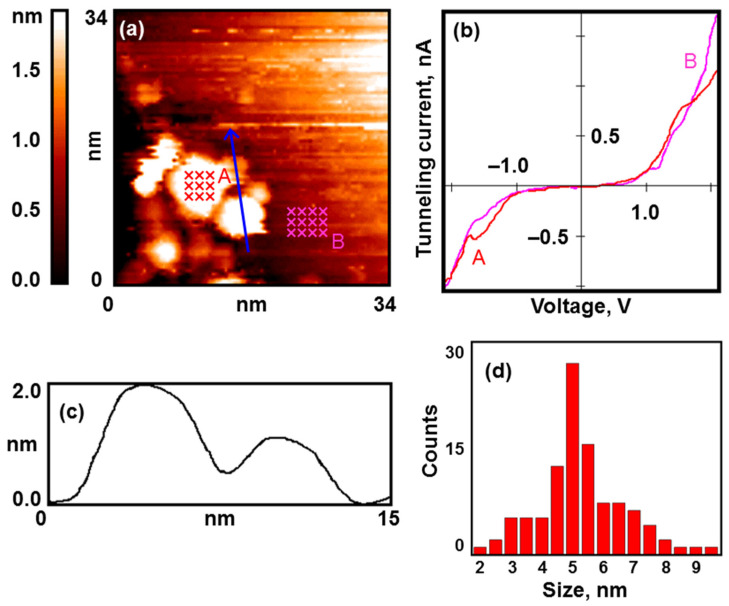
Nanoparticles of nickel after calcination under UHV conditions and reduction in H_2_. Results of the STM/STS measurement: (**a**) topography image of HOPG surface with deposited nickel nanoparticles; (**b**) VAC curves of the tunneling currents averaged over the set of points on the surface of nanoparticle (red curve A) and HOPG (pink curve B) marked with crosses in (**a**); (**c**) profile of the surface along the cut line shown in (**a**); (**d**) size distribution.

**Figure 6 nanomaterials-13-01365-f006:**
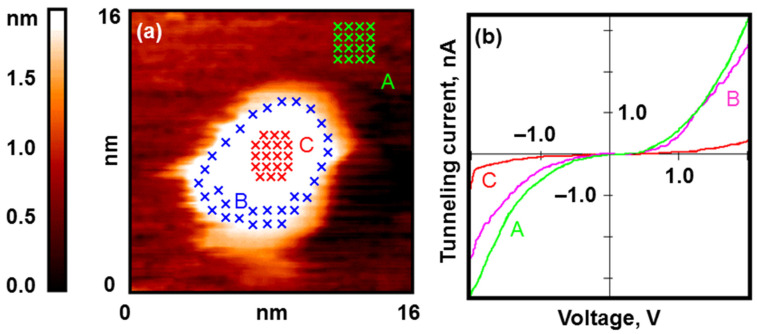
Nickel nanoparticles after exposure to O_2_ (200 L). Results of the STM/STS measurement: (**a**) topography image of the HOPG surface with deposited nickel nanoparticle with points of spectroscopy measurements marked with crosses; (**b**) VAC curves of the tunneling currents averaged over the set of points on the surface of the HOPG (green curve A), central (red curve C) and peripheral (pink curve B) areas of the nanoparticle marked in (**a**).

**Figure 7 nanomaterials-13-01365-f007:**
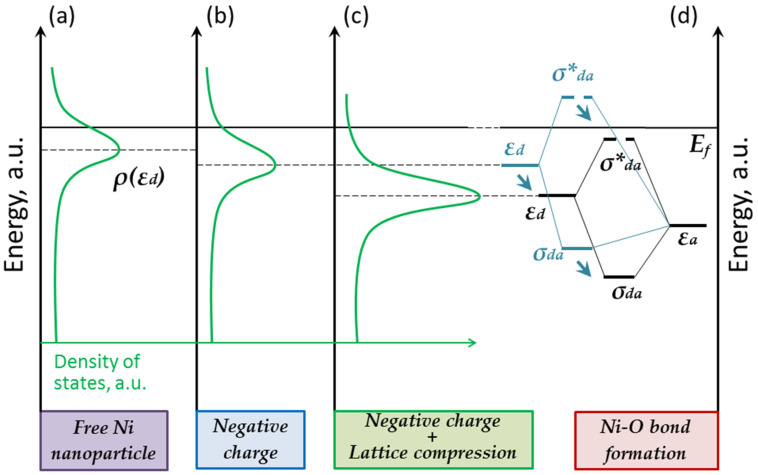
Influence of charge and deformation factors on the occupation of surface sp- and d-band states: (**a**) unsupported nickel nanoparticle; (**b**) negatively charged nickel nanoparticle; (**c**) nickel nanoparticle with excessive negative charge and compressed atomic lattice; (**d**) formation of bonding and antibonding orbitals due to the interaction of adatom with the sp-state and d-band state. The effective down-shift of the molecular orbitals after lattice compression is shown with blue arrows.

**Figure 8 nanomaterials-13-01365-f008:**
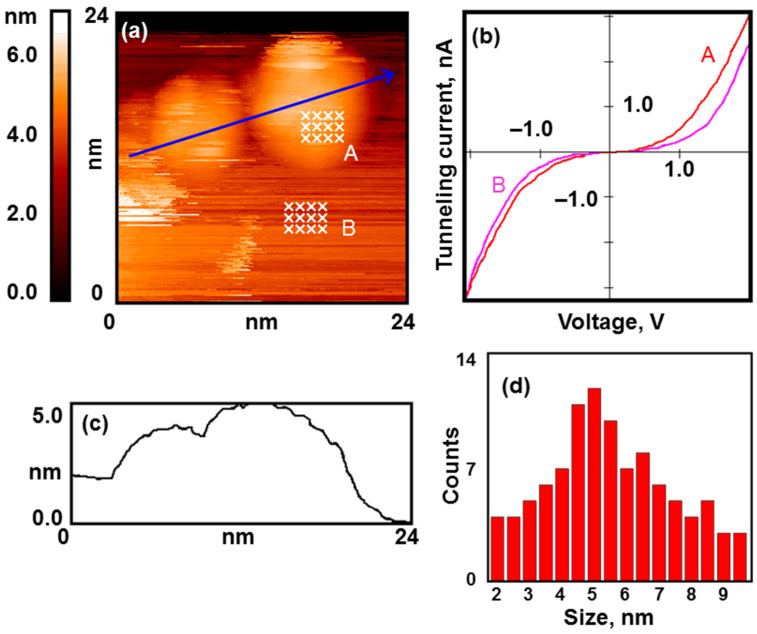
Nanoparticles of platinum after calcination under UHV conditions and reduction in H_2_. Results of the STM/STS measurement: (**a**) topography image of the HOPG surface with deposited platinum nanoparticles; (**b**) VAC curves of the tunneling currents averaged over the set of points on the surface of nanoparticles (red curve A) and HOPG (pink curve B) marked with crosses in (**a**); (**c**) profile of the surface along the cut line shown in (**a**); (**d**) size distribution.

**Figure 9 nanomaterials-13-01365-f009:**
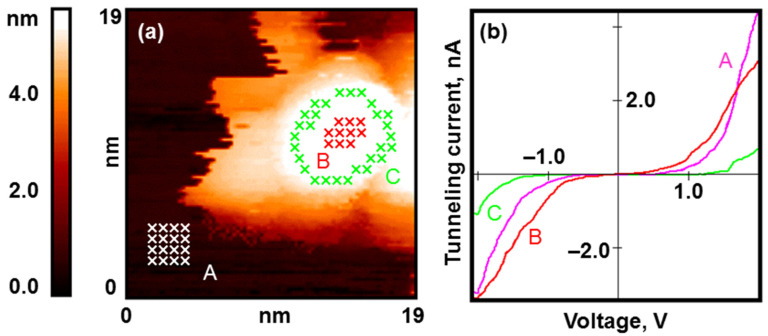
Oxidized platinum nanoparticles after exposure to H_2_ (200 L). Results of the STM/STS measurement: (**a**) topography image of the HOPG surface with deposited platinum nanoparticle with points of spectroscopy measurements marked with crosses; (**b**) VAC curves of the tunneling currents averaged over the set of points on the surface of HOPG (pink curve A), central (red curve B), and peripheral (green curve C) areas of the nanoparticle marked in (**a**).

**Figure 10 nanomaterials-13-01365-f010:**
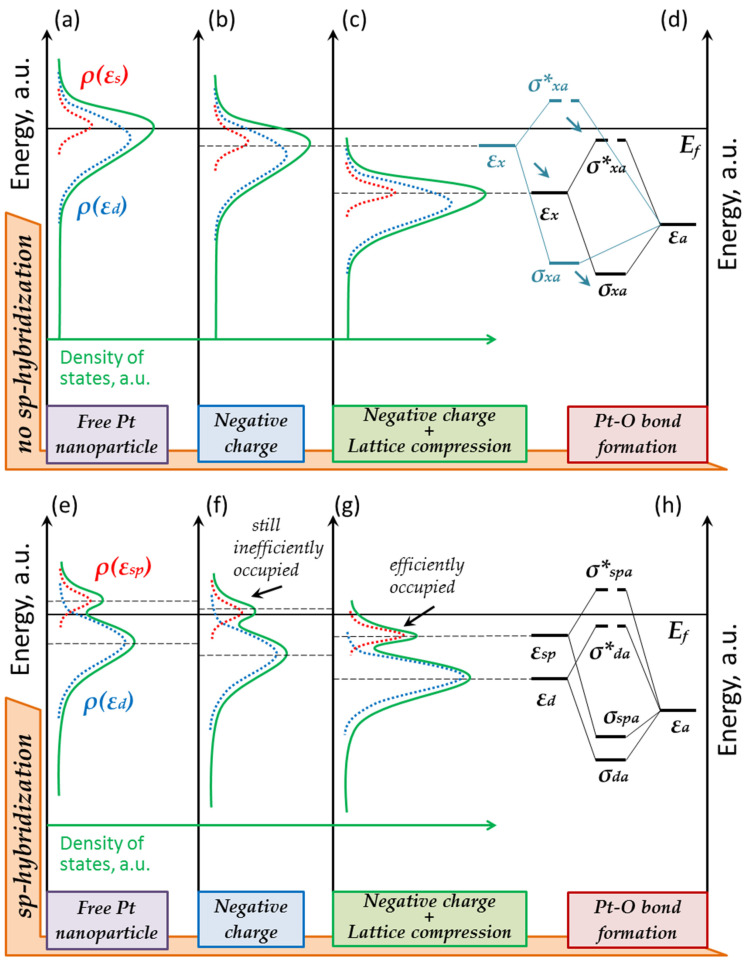
Adatom chemisorption on the surface of platinum nanoparticles without sp-hybridization (**a**–**d**) and with it (**e**,**f**). The influence of the charge and deformation factors is also shown: (**a**,**e**) unsupported platinum nanoparticle; (**b**,**f**) negatively charged platinum nanoparticle; (**c**,**g**) platinum nanoparticle with negative excessive charge and compressed atomic lattice; (**d**,**h**) formation of bonding and antibonding orbitals due to the interaction of adatom with surface states.

**Figure 11 nanomaterials-13-01365-f011:**
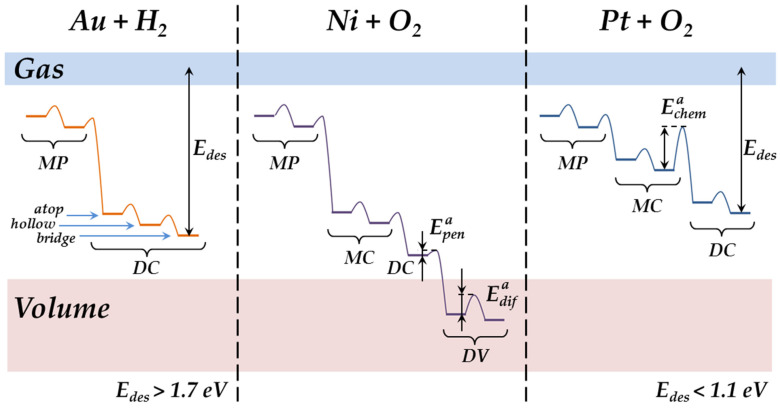
Simplified energy diagram of the processes taking place on the surfaces of gold, nickel, and platinum nanoparticles while interacting with gas molecules: MP—molecular physisorption, MC—molecular chemisorption, DC—dissociative chemisorption, DV—diffusion in volume.

**Table 1 nanomaterials-13-01365-t001:** Formation of the M-A_ads_ complex on the surface of bulk and nanostructured gold, platinum, and nickel. Differences in the adsorption properties for the upper and peripheral regions are shown for nanoparticles. Stable adsorption complex is marked with green tick, unstable—with red cross. Similarities in adsorption properties for different systems are shown with green filling.

.		M-A_ads_ Complex Formation		
		Au	Pt	Ni
Bulk		X	✓	✓
Nanoparticle on HOPG	Top	X	X	✓
	Periphery	✓	✓	X

## Data Availability

The data presented in this study are available on request from the corresponding author.

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
