# Peer review of "Less and Less Noble: Local Adsorption Properties of Supported Au, Ni, and Pt Nanoparticles"

_nanomaterials, 2023, doi:10.3390/nano13081365_

Round 1

Reviewer 1 Report

In this article, authors used STM and STS to study the gas absorption property of Au, Ni and Pt nanoparticles on HOPG surface. Contribution of surface states, lattice distortion and charging is discussed. This is an interesting work and I have a few questions below:

1. STM is a surface measurement and do not show the cross section information. However, the actual morphology of nano particle synthesized in this method (more hemispherical or more spherical) will matter especially for the contribution of lattice distortion. Can you provide additional result for the  nano particle-HOPG interface as well as nano particle size distribution?
2. As a follow up to question 1, is there any other mechanism that could lead to the semiconducting STS result near the nano particle surface other than gas absorption?
3. Since bulk and nano comparison is in the manuscript, could you please show case an absorption measurement with STS on bulk Au/Ni/Pt surface to directly see the size effect?
4. The lattice distortion in nanoparticles will depend on the reaction, the thermal expansion of the participating materials as well as the interaction between supporting surface and nano particle itself. Could you provide more evidence of lattice distortion and the %strain needed to cause the band shift needed?

Author Response

To Reviewer #1

Many thanks for Your valuable comments on our manuscript.

  1. You are absolutely right talking that lattice distortion will be affected by the shape of the nanoparticle. More spherical particle will have less contact area with the support than hemispherical, and so the distortion will be less significant. STM methods can’t provide us with the information about the interface structure — we just can’t watch under the external edge of the particle to establish the degree of the particle spherality. Yet we haven’t found appropriate method for such investigation. TEM is very sensitive for the lattice structure, but it gives only projected image of the material, so the partially distorted lattice can’t be resolved strictly. And XRD methods can’t deal with such low amount of material irregularly distributed over the HOPG surface. Methods of molecular dynamics and quantum-chemical simulation can help a bit, but deep energy optimization for the system with more than 1000 atoms costs too much time. Nevertheless we suppose the shape to be more spherical taking into account the high mobility of the particles over the HOPG surface. Low adhesion of gold and platinum to HOPG should result into the decrease in contact area of these materials. Information about nickel is ambiguous unfortunately. Of course this evidence is indirect, but the only thing we have. As for the size distribution the appropriate additions have been made to the article — see Figures 1,5,8.

  1. About the mechanism of semiconducting STS formation we can say, that all the investigated systems were stable before the injection of gas reagents in the experimental setup. Particles retain their size during the experiment, so we can discard such effects like size-dependent metal-semiconductor transition. So the only reason for significant decrease in conductance is strong localization of electrons caused by covalent bonds formation. Of course within the experiment we can’t distinguish adsorption, dissociation and adsorbate surface migration, but adsorption is the first step of any heterogeneous chemical reaction. So the gas adsorption is the essential part of all electronic structure changes we can observe.

  1. We have observed the size effect for all three systems and even managed to establish the threshold size. No hydrogen adsorption takes place on 10 nm Au nanoparticles (Nanomaterials 2019, 9(3), 344; https://doi.org/10.3390/nano9030344). No annular structure was observed for 15 nm Pt nanoparticles (unpublished). In the case of nickel this effect is anisotropic, since we have watched no oxide formation on thin planar nickel particles in disregard to their size (published partially in Nanomaterials 2022, 12(7), 1038; https://doi.org/10.3390/nano12071038 — see particles with mantel). But we would like to consider these results in further work since the correlation with mechanical features of these metals should be investigated precisely.

  1. We don’t think exothermic effect should be considered since HOPG is very good in heat conduction. This is one of the reason for that we use it. As for the distortion estimations — they are difficult. Of course we can’t investigate the structure of HOPG-Me interface. But we suppose the abovementioned result for planar nickel nanoparticles to be the evidence strong enough for changes in the geometry of 2-3 atomic layers of supported nickel. The most important evidence in the case of gold can be found in article (http://www.issp.ac.ru/lhpp/PapersAntonov/42e.pdf), where formation of AuH was considered under the pressure of 5 Gbar. From this article one can estimate that the decrease of Au-Au bond length for 1.2 % is enough for ignition of hydrogen adsorption on the surface of bulk gold and its further dissolution in metal. We would like to discuss this fact in our next article.

We are very grateful for the attention to our work and the comments made. We are open for further discussion and cooperation.

Reviewer 2 Report

This paper, entitled Less and Less Noble: Local Adsorption Properties of Supported Au, Ni and Pt Nanoparticles, presents an interesting strategy to determine a stable adsorption complex M-Aads on the nanoparticles surface. Although considerable work has been performed, several points must be improved for the acceptance of this manuscript.

  1. Please clearly mention in the last paragraph of the introduction the importance of this work and what the novelty is.
  2. On page 2, please be careful when writing the subscript and superscript in the experimental section.
  3. Please expand the research by performing some computation and illustrating with pictures the mechanism of the adsorption. Use as reference : DOI : 1186/1752-153X-6-91 ; DOI: 10.1039/D2CP04299A
  4. It will be helpful to insert the M-Aads bond length in a table and compare with the existing results from the reference.

Based on these, I advise the authors to rectify the above-mentioned errors, and I hope to re-evaluate the revised manuscript.

Author Response

To Reviewer #2

Many thanks for Your valuable comments on our manuscript.

  1. According to Your suggestion we have extended a bit the last paragraph of the Introduction. We have emphasized the importance of models describing the properties of nanomaterials in the case when chemical properties of materials have changed significantly and can't be strictly described in terms of solid state physics, and ab initio quantum-chemical modeling is very complicated. The ability to establish a correlation between the chemical properties of massive and nanosized particles of metals is very perspective. The approach developed by us eliminates the problem of describing such systems. Without loss of generality, it can be applied to similar systems based on nanoparticles of both transition and non-transition metals.

  1. Yeah, we have noticed that all the subscripts and superscripts failed during the making up, but it is the Journal fault. We had to correct all of them again.

  1. The whole mechanism includes molecular adsorption, molecular migration and dissociation, formation of stable M-Aads complex, surface migration of the adsorption complex. The main problem is that we can't watch the dynamics of the processes. STM/STS methods allow us to compare the initial and final condition of the investigated system. Of course we have some assumptions about all the stages of the interaction of molecules with nanoparticle surface. For example, in our previous work we have calculated that the most optimal position of H atom on the surface of small gold cluster is between a pair of Au atoms - 'bridge' position (Doklady Physical Chemistry 2016, 470(1), 125 https://doi.org/10.1134/S0012501616090013). But this result was obtained for cluster of 13 atoms and not for nanoparticle of 1000 atoms. And the calculations are getting too complicated if we just try to take into account the molecular dissociation. As for the experiment, we can't observe pure molecular adsorption in STM/STS experiments, we can't distinguish between the stages of dissociation and M-Aads formation, and in the case of high degree of surface occupation we can’t observe changes related to Aads migration. Of course it still allows us to consider precisely the M-Aads formation, but it’s not enough to describe all the reaction mechanism.

  1. We agree that information about bond length can be useful in some cases. But there are two reasons for why we don't discuss it in our manuscript. The first one is that STM/STS methods don't provide information about M-Aads bond length. Of course we can estimate the desorption energy by heating the sample and checking its surface condition. But we can't watch it in dynamics and also the migration and thermal rearrangement of the nanoparticles may have significant effect on desorption. The second reason is that the bond length depends significantly on the nanoparticle charge. In our previous work this effect was investigated for gold cluster (Nanotechnologies in Russia 2016, 11(11–12), 735 https://doi.org/10.1134/S1995078016060070). Of course the bond length also depends on the amount of atoms in cluster and on the degree of occupation of the surface with adsorbed atoms, but in the case of the nanoparticle of 1000 atoms these two factors are not so significant. Since the excessive charge will vary for every nanoparticle, the information about bond length will be useless.

We are very grateful for the attention to our work and the comments made. We are open for further discussion and cooperation.

Reviewer 3 Report

The manuscript "Less and Less Noble: Local Adsorption Properties of Supported Au, Ni and Pt Nanoparticles" deals with some interesting aspects of local adsorption properties of gold, nickel, and platinum nanoparticles. A correlation was established between the chemical properties of massive and nanosized particles of these metals. The formation of a stable adsorption complex M-Aads on the nanoparticles surface was described. It was shown that the difference in local adsorption properties is caused by specific contributions of nanoparticle charging, deformation of its atomic lattice near the M-C interface, and hybridization of surface s- and p-states. The authors have nicely represented the work with interesting data and results. The analysis results are very interesting as well. I enjoyed reading this article and I do recommend this manuscript for publication after following revisions.

1.     I think the introduction on the necessity and implications of this study for research in nano particles need to be extended. Although they are not directly related to your system, they have some useful discussions on the topic can be found in the following articles. I suggest authors consult these articles in their revision:

-        Improvement of Alcaligenes sp.TB performance by Fe-Pd/multi-walled carbon nanotubes: Enriched denitrification pathways and accelerated electron transport. Bioresource technology, 327, 2021, 124785. doi: 10.1016/j.biortech.2021.124785

-        Ge-Doped Cobalt Oxide for Electrocatalytic and Photocatalytic Water Splitting. ACS Catalysis, 12(19), 2022, 12000-12013. doi: 10.1021/acscatal.2c03730

-        Study on Electrochemical Properties of Carbon Submicron Fibers Loaded with Cobalt-Ferro Alloy and Compounds. Crystals, 13(2), 2023, 282. doi: 10.3390/cryst13020282"

2.     Can you please elaborate if quantum confinement has any contribution to your results?

3.     Can you further elaborate how the size contributes to your discussions? For instance, what is the implications of decreasing size from 10 to 5 nm?  

4.     Authors have talk about structure rearrangements and variation in crystal structures. Shouldn’t these arguments be further supported with XRD analyses?

Author Response

To Reviewer #3

Many thanks for Your valuable comments on our manuscript.

  1. Many thanks for these valuable links. According to Your suggestion we have extended a bit the Introduction. We have emphasized the necessity of models describing the properties of nanomaterials in the case when chemical properties of materials have changed significantly and can't be strictly described in terms of solid state physics, and ab initio quantum-chemical modeling is very complicated. The ability to establish a correlation between the chemical properties of massive and nanosized particles of metals is very perspective. The approach developed by us eliminates the problem of describing such systems. Without loss of generality, it can be applied to similar systems based on nanoparticles of both transition and non-transition metals.

  1. Quantum confinement may contribute a bit and change the band gap of oxides with semiconductor electronic structure. But it isn’t crucial as our nanoparticles consist of ~1000 atoms. Surface defects contribution is more significant. But if we are talking about STS experiments with more complicated adsorption complexes, e.g. HOH or HCO, of course, we should consider quantum effects related to the localization of the tunneling electrons on molecules as this process would affect significantly the tunneling efficiency and features of molecular spectra experimentally observed.

  1. We have observed the size effect for all three systems and even managed to establish the threshold size. No hydrogen adsorption takes place on 10 nm Au nanoparticles (Nanomaterials 2019, 9(3), 344; https://doi.org/10.3390/nano9030344). No annular structures were observed for 15 nm Pt nanoparticles (unpublished). In the case of nickel this effect is anisotropic, since we have watched no oxide formation on thin planar nickel particles in disregard to their size (published partially in Nanomaterials 2022, 12(7), 1038; https://doi.org/10.3390/nano12071038 — see particles with mantel). But we would like to consider these results in further work since the correlation with mechanical features of these metals should be investigated precisely.

  1. As for the structure rearrangements and variation in crystal structures — it is very difficult to estimate them. Yet we haven’t found appropriate method to investigate the interface distortion. STM methods can’t provide us with the information about the interface structure — we just can’t watch under the external edge of the particle. TEM is very sensitive for the lattice structure, but it gives only projected image of the material, so the partially distorted lattice can’t be resolved strictly. And XRD methods can’t deal with such low amount of material irregularly distributed over the HOPG surface. Methods of molecular dynamics and quantum-chemical simulation can help a bit, but deep energy optimization for the system with more than 1000 atoms costs too much time. At the same time we can get some estimations of distortion from the article (http://www.issp.ac.ru/lhpp/PapersAntonov/42e.pdf), where formation of AuH was considered under the pressure of 5 Gbar. From this article we can estimate that the decrease of Au-Au bond length for 1.2 % is enough for ignition of hydrogen adsorption on the surface of bulk gold and its further dissolution in metal. We would like to discuss this fact in our next article.

We are very grateful for the attention to our work and the comments made. We are open for further discussion and cooperation.

Reviewer 4 Report

In the present manuscript the authors studied the  local adsorption properties of gold, nickel, and platinum nanoparticles towards gases to give information about the catalytic activity of these nanoaparticles. Although the concepts beyond the manuscript are interesting and the dealt topic could be very appealing for a wide readership, the quality of presentation, in my opinion is poor. Through the manuscript the authors take into account data obtained in previous work and this, together with other aspects, makes the manuscript not clear. In my opinion it can be re-considered for publication on Nanomaterials after substantial revision, trying to improve the understandability of the experimentals and results discussed.

Some minor issues:

- I suppose the authors chose graphite as inert support, this or the reason of their choice, should be stated in the manuscript

- Please check all chemical formula by putting as subscripts the numbers.

- Mass spectroscopy should be change in mass spectrometry,

Author Response

To Reviewer #4

Many thanks for Your valuable comments on our manuscript.

  1. You are absolutely right talking about chemical inertness of HOPG — we have added this information in the beginning of the Experimental. But it is not the only reason why we use it. It is also very important fact that its surface looks like vast atomically smooth C(0001) terraces — we have mentioned about that. And by the way the standard method of preparation of its surface (exfoliation) is very easy-to-use.
  2. Yeah, we have noticed that all the subscripts and superscripts failed during the making up, but it is the Journal fault. We had to correct all of them again.
  3. Thanks a lot. We have corrected this mistake and changed mass spectroscopy to mass spectrometry.

We agree that the manuscript should describe some aspects of previous work in more details since it covers results of our last 7 years. But at the same time we have to deal with the fact that lots of details from previous articles can make the novelty of the idea presented unclear. Also huge articles are inconvenient for reading. So we had to cut off some details and suggested the readers to find additional information in our previous works.

We are very grateful for the attention to our work and the comments made. We are open for further discussion and cooperation.

Round 2

Reviewer 1 Report

Authors have done a good job clarifying the ambiguity. I believe the manuscript can be accepted as is. 

Author Response

Many thanks. We appreciate Your suggestions and are very grateful for the interest to our manuscript.

Reviewer 2 Report

The paper was improved after the first revision, particularly the introduction part of it. The authors only partially responded to my suggestions. I still recommend the authors to describe the mechanism of the adsorption. Some informations as described in your previous paper regarding Au as a metal (https://doi.org/10.1134/S0012501616090013) and also recommended references : DOI : 1186/1752-153X-6-91 ; DOI: 10.1039/D2CP04299A.

I understand that the results  were obtained for cluster of 13 atoms and not for nanoparticle of 1000 atoms, but as you mentioned, "we have some assumptions about all the stages of the interaction of molecules with nanoparticle surface" . Please make pictures or graphs to illustrate the mechanism.

The same mechanism is adopted by the Ni and Pt as well?

Author Response

To Reviewer #2

Many thanks for Your valuable comments.

According to Your suggestion we added a new part to our manuscript — 6. Short glance at adsorption mechanism. Unfortunately, we still don’t have enough information, for example, about activation energies for some stages or charges and adsorption forms. So building a diagram to illustrate the reaction paths is still a very difficult task. But we have described shortly the regularities of gas molecules interaction with surface for all three nanostructured systems. As we hope, this part contains enough details about adsorption mechanisms and demonstrates what processes and effects we have managed to elucidate.

We appreciate Your suggestions and are very grateful for the interest to our manuscript. We are open for further discussion and cooperation.

Reviewer 3 Report

First of all I would like to thank authors for their reply on my comments. But, I was expecting more from the revision and I am not satisfied unfortunately. I am afraid I will have to reject this paper, given that there some fundamentally important issues not being properly addressed in the manuscript.

First of all, I do not see how the introduction is improved. Adding 3-4 generic lines is not what I had in mind.

Moreover, the crystal structure is a key issue in this manuscript and cannot be left for the "next submission". I disagree with you about XRD not being an appropriate technique, as XRD provides information from the lattice spacing. 

Also, if quantum confinement does not have a major contribution, this needs to justified and discussed in the manuscript. 

Author Response

To Reviewer #3

Many thanks for Your valuable comments.

  1. According to Your previous suggestion we have included Your references (9–11) as good examples of fine interplay of various factors influencing chemical activity of the nanostructured systems. Together with previous extending of the Introduction, we suppose that to be enough for clarifying the novelty and value of our work. Of course, many interesting facts can be mentioned since our experiments address the field where quantum chemistry have to be connected with solid state physics, but we would like not to overweight the Introduction.
  2. Let's consider the structural rearrangement. We agree that it is very important, since the lattice compression influence is shown to be significant. But we suppose the references mentioned in the manuscript to be enough to prove the fact of compression. Of course the direct experiment on measuring the compressed lattice parameters is very desired. But that doesn't mean that indirect evidence is not suitable. For example, our conclusions about gold lattice compression is based on the results of calculations by Hammer and Nørskov (https://doi.org/10.1016/S1381-1169(96)00348-2), where gold was shown to be compressed when applied on most metal supports. The second evidence is the experimental result of Antonov group (http://www.issp.ac.ru/lhpp/PapersAntonov/42e.pdf), who managed to synthesize stable AuH and showed the interaction of bulk Au and H2 to be stimulated by compression of gold. As in our experiments we observe formation of stable Au-H product with semiconductor structure we also may conclude that the reason is gold lattice compression. And the third evidence is the experiment of Mays (https://doi.org/10.1016/0039-6028(68)90119-2), who managed to observe lattice compression for gold nanoparticles applied on carbon support by electron diffraction and transmission electron microscopy. Of course, that work dealt with surface tension, according to the authors, but it was discussible whether the carbon support influenced or not. Despite of that unclarity, they've managed to detect the decrease in interatomic distance of 0.3–0.4% for Au-C nanostructured system. We suppose these indirect evidences to be enough to state that lattice compression in the vicinity of Au-C interface is the most probable result of Au-C interaction, while direct experimental measuring of compressed lattice parameters is still impossible. The situation is the same if we consider Ni and Pt nanoparticles.

Now let's discuss the problem of direct measurement and Your suggestion to use XRD method to establish the compressed lattice parameters. Unfortunately there are some reasons that make XRD useless in this case. The main problem is the inhomogeneity of the samples. We have a mixture of nanoparticles of various sizes on the HOPG surface. In the case of gold the lateral size of the nanoparticles is 4–6 nm. There is a good example of XRD spectrum for carbon-glass supported gold nanoparticles in this work (https://doi.org/10.1039/B821016K), so we can consider how it will change in our case. The most intense peak is observed at 2θ=38.10° which corresponds to Au(111) surface. According to Scherrer equation one can calculate that the line broadening will be 4.67° for nanoparticles of 4 nm in diameter. For the nanoparticles of 6 nm the same parameter will be a bit smaller. According to our calculation the decrease in Au-Au bond length should be of 1.2% to stimulate the Au-H formation efficiently. So according to the Bregg’s law the new peak will be observed at 2θ=38.78° for compressed lattice, and the line broadening will be 3.11° for nanoparticles of 6 nm in diameter. Since the lattice compression is partial and occurs only in the vicinity of Au-C interface, the XRD spectrum will contain both peaks from standard and compressed lattice. And here is the problem again: it is impossible to resolve two peaks in the case when the difference in their position is 5–6 times less than the line broadening. And if we take into account that the vertical size of the nanoparticles is about 2 nm (and even less), the question arises: on what reason can we claim that XRD is suitable when the vertical size of the nanoparticle does not exceed 8–10 atomic layers? There is no need even to discuss the complicated methods of peak shape analysis in this case.

  1. About quantum confinement we still can say that De Broglie wavelength for electron in metal is about 1 Å, and the nanoparticle size is 50 times bigger. Nanoparticles are big enough to possess standard band structure. If it were not so, we might have observed distant peaks in STS experiments instead of smooth S-shaped VAC curve. In the case of oxidized nanoparticles the quantum confinement maybe could somehow lead to the increase in the band gap value, but in our experiments we also don't observe it — all the band gaps correlate with reference values for bulk oxides or even decrease due to the deviations in stoichiometry. These are quite obvious reasons why we don't consider the quantum confinement and use Newns-Anderson chemisorption model. We don't see any necessity to discuss this in details in the manuscript.

We appreciate Your suggestions and are very grateful for the interest to our work and for comments. We are open for further discussion and cooperation.

Reviewer 4 Report

The authors revised the manuscript accordingly to my suggestions and where not they provide convincing explanation of their choice, thus in my opinion it can be now be accepted for publication on Nanomaterials.

Author Response

(The authors gave the same response as above.)

Round 3

Reviewer 2 Report

The paper was improved after the second revision, particularly the introduction part of it, and as recommended, the mechanism was slightly discussed. I still consider that a graphical representation will better describe the mechanism. Some information as described in your previous paper regarding Au as a metal (https://doi.org/10.1134/S0012501616090013) was introduced, but it seems that the recommended references were not cited in your paper : DOI : 1186/1752-153X-6-91 ; DOI: 10.1039/D2CP04299A. Adjust this annotation and insert the recommended reference where you describe the mechanism.

Author Response

Many thanks for Your suggestion that helped us to improve the manuscript significantly. We have added the diagram describing the main stages of gas molecules interaction with gold, nickel and platinum nanoparticles (see Figure 11). 

Reviewer 3 Report

I would like to thank authors for very extensive and convincing explanations in this round of revision. I enjoyed discussions we had and I suggest this revised version for publication. Also, I look forward to the next paper from this group with more focus on details we discussed about in this paper. 

Author Response

Many thanks for Your questions and valuable comments that helped us to improve the manuscript. We enjoyed the discussions too.